# Effects of Swirl and Boiling Heat Transfer on the Performance Enhancement and Emission Reduction for a Medium Diesel Engine Fueled with Biodiesel

**Dongli Tan [1,2], Zhiyong Chen [3], Jiangtao Li [1,2], Jianbin Luo [1,2,*], Dayong Yang [1], Shuwan Cui [1] and Zhiqing Zhang [1,2]**

1  School of Mechanical and Transportation, Guangxi University of Science and Technology, Liuzhou 545006, China; istandongli@gxust.edu.cn (D.T.); 221055037@stdmail.gxust.edu.cn (J.L.); dyyang@gxust.edu.cn (D.Y.); 13597066615@163.com (S.C.); zhangzhiqing@gxust.edu.cn (Z.Z.)
2  Institute of the New Energy and Energy-Saving & Emission-Reduction, Guangxi University of Science and Technology, Liuzhou 545006, China
3  Department of Numerical Control Technology, Guangxi Mechanical and Electrical Technician College, Liuzhou 545005, China; gxgjxb@126.com
*  Correspondence: luojianbin@gxust.edu.cn

**Abstract:** In order to improve the accuracy of numerical simulation, a new heat transfer model is developed by using a modular approach in the Anstalt für Verbrennungskraftmaschinen (AVL)-Boost software. The improved heat transfer model mainly considers the effects of the swirl and boiling heat transfer inside the engine. In addition, a chemical kinetics mechanism including 475 reactions and 134 species is employed to predict the combustion of diesel engines fueled with biodiesel. The result shows that the boiling heat transfer will occur, especially in the high-temperature area. Analysis shows that the improved model is reliable and its precision is increased. Finally, the perturbation method is employed to investigate the relatively important inputs as the complex nonlinear function with a lot of output data and input data produced by the improved model. The relative effects of different parameters such as *EGR*, injection mass, injection timing, compression ratio, inlet air pressure, fuel injection pressure, exhaust pressure and inlet air temperature on performance and emission characteristics are compared. The eight parameters are investigated on four outputs of brake power, Brake Specific Fuel Consumption (*BSFC*), $NO_x$ and HC. The injected fuel mass plays an important role in emissions and performance. The *EGR*, compression ratio and inlet air pressure have a great effect on the HC and $NO_x$ emission.

**Keywords:** heat transfer model; perturbation sensitivity analysis; performance and emission characteristics; diesel engine

## 1. Introduction

Due to the high economy and reliability of diesel engines, they are used as the main power source and widely used in the ship, construction machinery and heavy-duty trucks [1,2]. With the increase in diesel engines, the particulate matter (PM) and $NO_x$ emissions from diesel engines have posed a threat to our health and ecosystem [3]. The international community is paying more and more attention to it. At the same time, the demand for engine performance and emissions is stronger and stronger. Thus, it requires us to investigate the knowledge of in-cylinder phenomena including chemical kinetics that determine the formation of different emissions, spatially-varying and temporally in-cylinder charge conditions, *EGR*, heat release rates, selective catalyst reduction and so on [4]. However, the new technology makes it more complex to adjust all input parameters [5]. It is necessary to further optimize the performance and emission characteristics of diesel engines with the new technologies [6,7].

The diesel engine can convert chemical energy into thermal energy, mechanical energy or other energy to meet their respective demands. Many scholars and engine manufactures have found that biodiesel fuel is an environment-friendly fuel [8]. However, there are significant differences in physicochemical properties, atomization and combustion characteristics between biodiesel and diesel [7]. It needs thorough theory and experimental study when biodiesel has been widely applied and realized highly effective and pure burning in the engine [9]. In addition, improvements in diesel energy efficiency and emission can be obtained in some areas of design and operation [10]. Meanwhile, many scholars and engine manufactures have developed many methods to reduce the emission and improve engine efficiency [11]. For instance, the exhaust valve is controlled by computer-controlled high-pressure hydraulic systems [12]. In addition, the waste heat recovery system [13], *EGR* and selective catalytic reduction technologies are employed for the diesel engine [14]. Nevertheless, many possible configurations for diesel engines require a shorter engine start-up time [15]. Thus, a numerical prediction model should be employed to predict the performance of diesel engines and the optimal configuration of the engine was obtained [16]. However, it is generally believed that the ideal model cannot fully and accurately reflect the characteristics of the current internal combustion engine due to its inherent limitations. For instance, the ideal model ignores the influence of many secondary considerations [17,18].

The engine prototyping and experiment in manufacturing industries in the design processes will cost a lot of money and time [19]. Thus, many scholars have carried out research on numerical simulation [20]. The cooling water system has an effect on heat transfer efficiency and combustion efficiency [21]. Previous research has focused on the development of the heat transfer model. In general, the boiling phenomena may occur in the hot spots of diesel engines due to the increasing temperature. If the boiling heat transfer in the hot spots of the diesel engine is not considered, about 33% of fuel energy is converted to heat loss from the cylinder body for large-scale diesel engines [22]. Reference [23] about 50% of fuel energy is transformed into heat loss for small-scale diesel engines. Thus, it is necessary to consider the boiling heat transfer due to the heat transfer characteristics of diesel engines. For instance, a validated single-phase boiling model had been developed by Gils et al. and was employed to investigate the boiling heat transfer of diesel engines [24]. They found that the calculation accuracy was improved when the model considered the boiling heat transfer of diesel engines. In addition, the two-phase boiling model of heat transfer coefficient had been developed by Mohummadi et al. and they found that the heat transfer coefficient obtained by the single-phase boiling model was close to the results obtained by the mixed two-phase flow analysis [25]. To better evaluate the heat transfer state of engine parts, the boiling heat transfer in the cylinder should be considered.

In order to solve this problem, the numerical method is considered an effective way to evaluate the practicability of biodiesel and it is widely used as a tool for the improvement and alternatives in the process of engine design with the limited resources [26]. However, the computational time and accuracy of the numerical model are important problems [27]. Only to ensure the error accuracy in an acceptable range can better evaluate the heat transfer state of engine parts such as cylinder liner and piston and meet the engine design and development process [28]. The swirl and boiling heat transfer in diesel engines should be considered [29]. In the hot spots of the engine, the swirl and heat transfer of nucleate boiling often occur especially in high load operation [30]. In general, the boiling heat transfer can be divided into two parts: saturated boiling and sub-cooled [31]. The film boiling on the surface tubes will result in a significant change in heat transfer and the heat flux from surface to cooling water will be reduced suddenly [32]. Thus, the surface temperature will increase rapidly and damage the engine if the temperature is too high [33]. Similarly, the swirl will occur in the combustion chamber. The zero-dimensional model can be widely employed to evaluate both the transient response and steady-state performance evaluation, in cases where the requirements for predicting results are limited. If both the issues are factored in, the given model is accurate [34].

As previously described, it needs thorough theory and experimental study when biodiesel is widely applied and realized highly effective and pure burning in the engine, because there are differences between the biodiesel and diesel in the physics and chemistry characteristic, as well as the atomization characteristic and the combustion characteristic. In the paper, an improved heat transfer model is proposed based on the basic theory and numerical method of swirl and boiling heat transfer. In order to evaluate the practicability of biodiesel, the model combined with Chemkin code is employed to simulate the cylinder combustion process of diesel engines. The experiment is carried out with a medium-speed diesel engine and the improved model is validated under the different engine conditions. Moreover, the effects of the swirl and boiling heat transfer on the combustion and emission characteristics of diesel engines fueled with biodiesel are investigated and compared. Ultimately, the relative influences of input parameters on combustion and emission characteristics are investigated based on the perturbation theory.

## 2. Methods and Model Validation

To simplify the simulation process, the inlet and exhaust are considered as the ideal gas. The main mathematical models are as follows.

### 2.1. Mathematic Modeling

The entire diesel engine layout featuring including four cylinders a detailed description of the flow path is shown in Figure 1. The air is cooled by the air cooler and enters the engine cylinder through the air intake passage. The exhaust gases are discharged and flow into the turbine inlet in the exhaust stroke. The turbine drives the turbo-compressor which delivers the compressed air to the cooler.

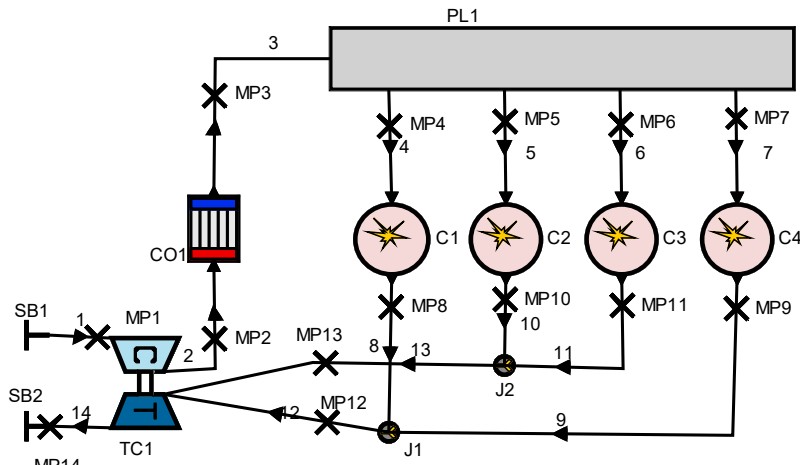

**Figure 1.** Simulation model of the entire diesel engine.

(1)   Energy Balance

The mass and energy balance problem caused by fuel injection and blow-by is solved between the inlet valve close and the exhaust valve open. According to the First Law of Thermodynamics, the system can be regarded [35], and the expression of the open system can be expressed as:

$$\mathrm{d}U_c = -p_c\mathrm{d}V - \mathrm{d}Q + h_{f,inj}\mathrm{d}m_f - h_{BB}\mathrm{d}m_{BB} \tag{1}$$

where $p_c\mathrm{d}V$ is the expansion work, J; $Q$ is the heat transfer to the piston, cylinder liner and cylinder head, J; $m_f$ is the injected fuel mass, kg; $h_{f,inj}$ is the specific enthalpy of fuel, J/kg; $m_{BB}$ is the air blow-by mass, kg; $h_{BB}$ is the specific enthalpy of air blow-by, J/kg; $Uc$ is the variation of internal energy, J.

(2)   Mass Balance

The leakage of gas mass results from the addition of residual gas mass and minus the short-circuit mass, fresh air mass, the exhaust gas recirculation mass during valve overlap. The fresh air mass and exhaust gas recirculation mass are obtained from experimental measurement. The residual gas mass and the short-circuit mass can be calculated by an emptying and simple filling model [10]. Thus, the mass fraction at *EVO* can be calculated as: When the exhaust valve opens, the mass fraction can be calculated as:

$$Y_{b,EVO} = \frac{m_f + \frac{m_f}{F_s} + (m_{EGR} + m_{res}) \cdot Y_{b,EVO} - m_{sc} \cdot Y_{b,IVC} - m_{BB}\left(\frac{Y_{b,IVC} + Y_{b,EVO}}{2}\right)}{m_a + m_f + m_{EGR} + m_{res} - m_{sc} - m_{BB}}$$
$$= \frac{1 + \frac{1}{F_s} - Y_{b,IVC}\left(\frac{m_{sc} + m_{BB}/2}{m_f}\right)}{1 + \frac{1}{F} - \frac{m_{sc} + m_{BB}/2}{m_f}}$$

(2)

where $Y_{b,EVO}$ is the mass fraction at exhaust valve open, %; $F_s$ is the stoichiometric fuel–air equivalence ratio; $m_{res}$ is the residual gas for the previous cycle, kg; $m_a$ is the fresh air, kg; $m_{sc}$ is the short-circuit mass, kg; $m_{EGR}$ is the exhaust gas recirculation mass, kg; $m_{res}$ is the residual gas for the previous cycle, kg.

Similarly, the mean value of compositions is calculated for the blow-by the leakage of gas mass [11]. The composition at inlet valve close can be expressed as:

$$Y_{b,IVC} = \frac{(m_{res} + m_{EGR}) \cdot Y_{b,EVO} - m_{sc} \cdot Y_{b,IVC}}{m_a + m_{res} + m_{EGR} - m_{sc}} = \frac{Y_{b,EVO}}{1 + \frac{m_a}{m_{EGR} + m_{res}}}$$

(3)

Thus, the mass fraction [11] at exhaust valve open is expressed as:

$$Y_{b,EVO} = \frac{1 + \frac{1}{F_s}}{1 + \frac{1}{F} - \left(\frac{m_{sc} + m_{BB}/2}{m_f}\right)\frac{1}{1 + \frac{m_{EGR} + m_{res}}{m_a}}}$$

(4)

(3)   Blow-by Model

The scavenging mass flow rate is calculated according to the following equation, which was derived assuming subsonic flow consideration through the valves [36] during the valve overlapping period:

$$\dot{m}_{BB} = C_d A_{ref} \frac{p}{\sqrt{RT}} \sqrt{\frac{2\gamma}{\gamma - 1}\left[\left(\frac{p_{crk}}{p}\right)^{\frac{2}{\gamma}} - \left(\frac{p_{crk}}{p}\right)^{\frac{\gamma + 1}{\gamma}}\right]}$$

(5)

where $C_d$ is the discharge coefficient; $A_{ref}$ is the reference section, m$^2$; $p_{crk}$ is the crankcase pressure, Pa; $\gamma$ is the adiabatic index.

To ensure the simulation result in accord with the actual cumulative blow-by, $C_d$ is should be adjusted with experimental results [10]. In addition, the flow area of equivalent cylinders is calculated by the following equation [36]:

$$A_{ref} = \frac{z_{cyl}}{\Delta\varphi} \int_0^{\Delta\varphi} \frac{A_{IVC(\varphi)} A_{EVO(\varphi)}}{\sqrt{A_{IVC(\varphi)}^2 + A_{EVO(\varphi)}^2}} d\varphi$$

(6)

where $Z_{cyl}$ is the number of engine cylinders; $\varphi$ is the crank angle, °.

(4)   AVL MCC Combustion Model

The quasi-dimensional MCC AVL combustion model in the AVL-BOOST environment, which considers the effects of the working fluid pre-mixing and diffusion combustion, is employed to predict the combustion in the cylinder [37]. The expression is expressed as:

$$\frac{dQ_F}{d\varphi} = \frac{dQ_{MCC}}{d\varphi} + \frac{dQ_{PMC}}{d\varphi} \tag{7}$$

where $Q_F$ is the heat release rate, J; $Q_{MCC}$ is the heat release rate in the diffusion combustion process, J; $Q_{PMC}$ is the heat release rate in the pre-mixed combustion process, J.

The expression of heat release rate [38] in the diffusion combustion process is expressed as:

$$\frac{dQ_{MCC}}{d\varphi} = C_{comb} \cdot \left( m_F - \frac{Q_{MCC}}{H_u} \right) \cdot (w_{air})^{C_{EGR}} \cdot C_{Rate} \cdot \frac{\sqrt{k}}{\sqrt[3]{V}} \tag{8}$$

where $m_F$ is the fuel evaporative mass, kg; $H_u$ is the low calorific value, J/kg; $w_{air}$ is the effective air mass fraction,%; $k$ is the turbulent energy density, kg/(m·s²); $C_{comb}$ is the combustion constant; $C_{EGR}$ is the combustion constant; $C_{Rate}$ is the constant of mixing ratio.

The expression of heat release rate [39] in the pre-mixed combustion process is expressed as:

$$\frac{1}{Q_{PMC}} \frac{dQ_{PMC}}{d\varphi} = \frac{6.908}{\Delta\varphi_C} \cdot (m+1) \cdot \left( \frac{\varphi - \varphi_B}{\Delta\varphi_C} \right)^m \cdot \exp\left[ -6.908 \cdot \left( \frac{\varphi - \varphi_B}{\Delta\varphi_C} \right)^{(m+1)} \right] \tag{9}$$

where $\varphi_B$ is the start of combustion angle, °; $\Delta\varphi_C$ is the difference of crank angle, °; $m$ is the shape parameter.

(5)   Heat Transfer Model

The heat transfer in the cylinder of the diesel engine is a very complicated process. The convection heat transfer is obtained from the Woschni 1978 heat transfer model [18]. The Woschni 1978 heat transfer model was published in 1978. It can be expressed as:

$$\alpha_w = aT_c^{-0.53} p_c^{0.8} D^{-0.2} \left[ C_1 C_m + C_2 \frac{V T_{IVC}}{p_{IVC} V_{IVC}} (p_c - p_{c,o}) \right]^{0.8} \tag{10}$$

where $\alpha_w$ is the heat transfer rate, W/(m·K); $T_c$ is the engine cylinder temperature, K; $C_m$ is the mean piston velocity, m²; $p_c$ is the cylinder pressure, Pa; $D$ is the cylinder diameter, m; $T_{IVC}$ is the cylinder temperature when the inlet valve closes, K; $V$ is the engine cylinder volume, m³; $p_{c,o}$ is the inverted cylinder pressure, Pa; $p_{IVC}$ is the cylinder pressure when the inlet valve closes, Pa; $a$ is a curve fitted constant; $C_1$ is the gas velocity coefficient; $C_2$ is the model constant.

In general, five typical states are defined by the existing heat transfer characteristics and conditions: convection, developed nucleate boiling, fully developed nucleate boiling, transition boiling and film boiling. It can be found that the saturation temperature of the water is not constant and changes with pressure in the engine. The difference between the wall temperature and the water saturation temperature $\Delta T$ is the wall superheat. When $\Delta T < 4$ K, the superheated water vaporizes when it reaches the free surface. With the increase in wall superheat, a large number of bubbles produced near the wall regime will escape from the wall into the surrounding liquid. Because the mainstream liquid is still in the supercooled state, it will condense and disappear. In the nucleate boiling state, the bubbles are continuously produced and destroyed, resulting in severe disturbances. The heat transfer efficiency is improved and the heat flux increases dramatically. When the superheat reaches $\Delta T_D$, the maximum value of heat flux is called critical heat flux. When $\Delta T_D < \Delta T$, the heat flux decreased dramatically. Thus, it is very important that the boiling heat transfer is controlled within the nucleate boiling state. In the present paper, the heat transfer model considers the influence of the swirl and the boiling heat transfer of the engine. The boiling of cooling water inside the engine is subcooled flow boiling [11]. The

total heat on the wall surface includes the phase transformation latent heat and the increase in the cooling water temperature [26]. The total heat transfer $q_{total}$ consists of boiling heat transfer and convective heat transfer. The $q_{total}$ is expressed as:

$$q_{total} = q_c + q_B \tag{11}$$

where $q_B$ is the sub-cooling boiling heat transfer heat flux, W/m$^2$; $q_c$ is the convective heat flux, W/m$^2$.

The single-phase convective heat flux $q_c$ should be expressed as:

$$q_c = \alpha_c(T_w - T_f) \tag{12}$$

where $\alpha_c$ is the single-phase convective heat transfer coefficient.

The improved heat transfer model was proposed based on the Woschni1978 heat transfer model. It considers the swirl and boiling heat transfer in the cylinder. These improvements are performed by means of AVL-Boost software, as it is described in the references [11]. The heat transfer coefficient should be expressed as:

$$\alpha_c = bT_c^{-0.53}p_c^{0.8}D^{-0.2}\left[C_{w1}C_m + C_uC_{w2} + C_{2u}\frac{VT_{IVC}}{p_{IVC}V_{IVC}}(p_c - p_{c,o})\right]^{0.8} \tag{13}$$

where $b = 0.12$ and $C_{2u} = 0.001$, $C_u$ is the instantaneous tangential gas velocity in the cylinder, $C_{w1}$ and $C_{w2}$ are fitted constants whose values are obtained from motoring tests on a specific diesel engine.

In Incropera's opinion [40], the boiling heat transfer coefficient can be calculated by the experimental correlations and the boiling heat transfer heat flux is summarized as follows:

$$q_B = \mu_L r\sqrt{\frac{g(\rho_L - \rho_g)}{\delta}}\left(\frac{c_{pL}\Delta t}{c_{wL}r\mathrm{Pr}_L^{1.7}}\right)^3 \tag{14}$$

where $\delta$ is the surface tension of the gas–liquid interface, $10^{-3}$ N/m; $\Delta t$ is the wall superheat degree, K; $r$ is the liquid vaporization latent heat, J/kg; $\mathrm{Pr}_L$ is the Prandtl number; $c_{pL}$ is the specific heat capacity of saturated liquid, J/(kg·K); $c_{wL}$ is the combinative empirical coefficient of the liquid and heating surface.

(6)  Emission Models

NO is mainly produced in the combustion process of diesel engines. The formation mainly includes the following three ways. (1) NO is generated in the high-temperature region; (2) NO is generated immediately when the fuel is ignited in the cylinder; (3) NO is produced by the nitrogen contained in the fuel. Thus, the extended Zeldovich model is employed to predict the NO$_x$ emission [41]. Similarly, the self-contained CO mechanism of the Zeldovich model is employed to predict the CO emission [41]. In addition, the Frolov Kinetic Model is employed to predict the HC emission [41]. The model mainly includes the simplified mechanisms of HC formation and HC oxidation.

### 2.2. Fuel Samples Preparation

The alkali catalysis was employed for the transesterification of rapeseed oil with methanol to produce the rapeseed oil methyl ester (RME) in a reactor about 1 h. RME is composed of five parts based on the fatty acid profile, including methyl linolenate (C18:3), methyl linoleate (C18:2), methyl oleate (C18:1), methyl stearate (C18:0) and methyl palmitate (C16:0). Cm:n is the shorthand of fatty acid methyl esters. m and n are the numbers of carbon atoms and double bonds, respectively. The physical properties of RME are shown in Table 1. The detailed information on RME can be found in our team's previous work [8,14,27].

**Table 1.** Properties and Composition of rapeseed oil methyl ester (RME).

| Item | Value | Item | Value |
|---|---|---|---|
| Lower calorific value (MJ/kg) | 39.53 | Oxygen content (% m/m) | 10.7 |
| Viscosity at 40 °C (mm·s$^{-2}$) | 4.56 | Cetane number (–) | 53.88 |
| Surface tension (dynes·cm$^{-2}$ at 25 °C) | 30.66 | Saturation (%) | 4.45 |
| Methyl linoleate (%) | 8.11 | Methyl stearate (%) | 65.18 |
| Methyl linoleate (%) | 22.27 | Methyl palmitate (%) | 0.87 |
| Methyl palmitate (%) | 3.57 | | |

### 2.3. Feasibility Test

The hydraulic dynamometer is coupled to a diesel engine and the experiment is carried out in the test room. The schematic of the experimental device is shown in Figure 2. An AVL DEWE-2010CA is employed to monitor the combustion in the cylinder. A Xiangyi eddy current dynamometer is used to measure the diesel engine load. Similarly, a Horiba MEXA-1600 type metes and an AVL Dismoke-4000 are used to measure the NO$_x$ and soot emissions, respectively. In addition, the appropriate sensors are employed to measure the flow, temperature and pressure.

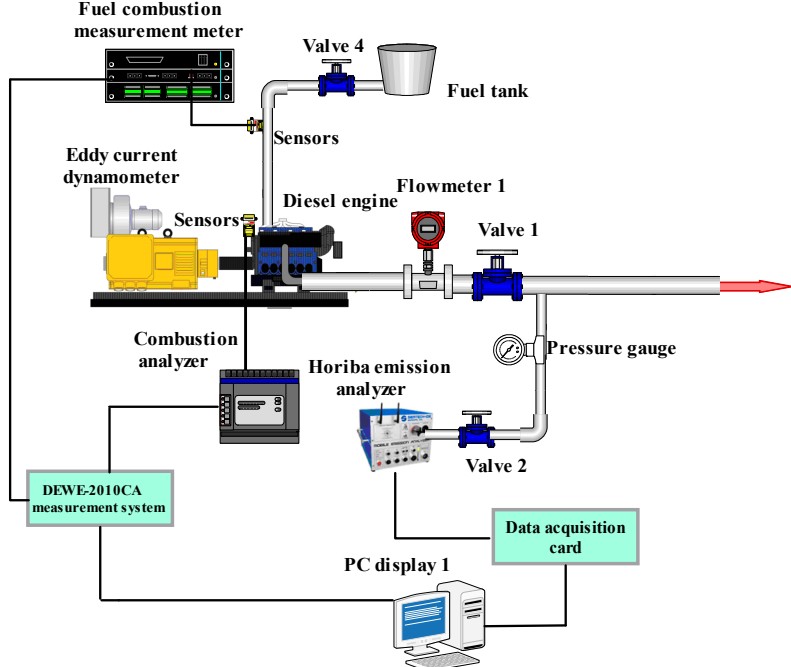

**Figure 2.** Schematics of experimental device.

### 2.4. Model Validation

In order to validate the improved heat transfer model, the experiment was carried out with a four cylinders diesel engine. The following input data are employed to develop the model: the equivalent area of the exhaust and intake valves, the engine geometric data, the steady-state compressor, the ambient conditions and so on. Then the model of initial boundary conditions and input parameters of the basic parameters are shown in Table 2. The adjustment process of the sub-model was corrected and the global model validation was carried out. Thus, the experimental measurements were performed in a medium-speed direct-injection diesel with four cylinders.

**Table 2.** Engine Specifications and Boundary Condition.

| Type | Value | Type | Value |
|---|---|---|---|
| Bore×stroke (mm) | 190 × 210 | Cylinder turbulence scale length (mm) | 3 |
| Number of cylinders | 4 | Initial pressure in the inlet (MPa) | 0.157 |
| Connecting rod (mm) | 410 | Initial temperature in the inlet (K) | 313.15 |
| Nozzle radius (mm) | 0.26 | Exhaust valve opening | 58° BBDC |
| Fuel injection holes | 8 | Exhaust valve closing | 56° ATDC |
| Cylinder diameter (mm) | 190 | Intake valve opening | 66° BTDC |
| Compression ratio | 14 | Intake valve closing | 54° ABDC |
| Initial cylinder turbulent kinetic energy (m$^2$/s$^2$) | 18.375 | | |

As previously described, the models described in the previous section have lots of constants, and the constants should be adjusted. It is necessary to set the constant values, especially the values of the heat transfer constant $C_{w1}$ and $C_{w2}$. The calibration of the model is done using experimental measurements. The values of $C_{w1}$ and $C_{w2}$ ae adjusted by using experimental motoring tests carried out on the diesel engine to be modelled. In motoring conditions, the adjustment method is based on the sensitivity of the thermodynamic cycle according to the Lapuerta et al. studies [42]. The heat transfer to the walls could be calculated by using the energy balance equation, which provided the heat transfer experimental measurement based on in-cylinder pressure. An equivalent analysis could be finished by imposing the improved heat transfer model considering boiling and swirl and the error of heat transfer could be obtained (theoretically the heat released in a calculation step is zero). Thus, it is possible to adjust the actual values of the uncertainties so that the engine experimental heat transfer to the walls agrees with the computational results of the improved heat transfer model. Thus, the error is minimized.

The medium-speed diesel engine is widely used for the propulsion of inland river ships. In addition, the engine speed and power exist the cubic relationship. In accordance with international IS08178 standards, the diesel engine was performed at the E3 test cycle mode. The propulsion characteristic of the diesel engine was investigated at four different speed output levels of 628 rpm, 799 rpm, 909 rpm and 1000 rpm respectively, corresponding to 25%, 50%, 75% and 100% load in the study. Thus, the characterization procedure is carried out on a set of motoring tests with the measuring range and accuracy of diesel parameters as detailed in Table 3. The results obtained from the fitting process are $C_{w1}$ = 1.82 and $C_{w1}$ = 1.12.

**Table 3.** Motoring Tests.

| Parameter | Units | 1 | 2 | 3 | 4 |
|---|---|---|---|---|---|
| Speed | (rpm) | 1000 | 909 | 799 | 628 |
| Inlet pressure | (bar) | 1.550 | 1.339 | 1.169 | 1.064 |
| Inlet temperature | (°C) | 44 | 41.9 | 40.9 | 40.4 |
| Exhaust pressure | (mbar) | 1750 | 1521 | 1460 | 1534 |
| Exhaust temperature | (°C) | 481 | 429 | 358 | 279 |

In the calculation processes, the improved model, which considers the swirl and boiling heat transfer, is used to calculate the heat transfer. In addition, a four-component chemical kinetics mechanism including 475 reactions and 134 species was developed in our team's previous work [14,27]. The mechanism consisting of methyl linolenate, methyl oleate, methyl stearate and methyl palmitate is used to calculate the cylinder combustion process in the paper. To validate the improved model, the experiments were carried out and the comparison of cylinder pressure at 1000 rpm under different load conditions were shown in Figure 3. It can be found that the improved model considering the swirl and boiling heat transfer can predict accurately the engine cylinder pressure compared with the Woschni 1978 model. For instance, the maximum error of the max. cylinder pressure

without considering the swirl and boiling heat transfer reaches 4.2%, but the maximum error of the max. cylinder pressure with the improved model is 2.1%. Thus, the improved model shows reasonable accuracy. This is due to the fact that the improved model considers the influence of the swirl and boiling heat transfer inside the engine. Thus, the agreement between predicted and measured cylinder pressure is better.

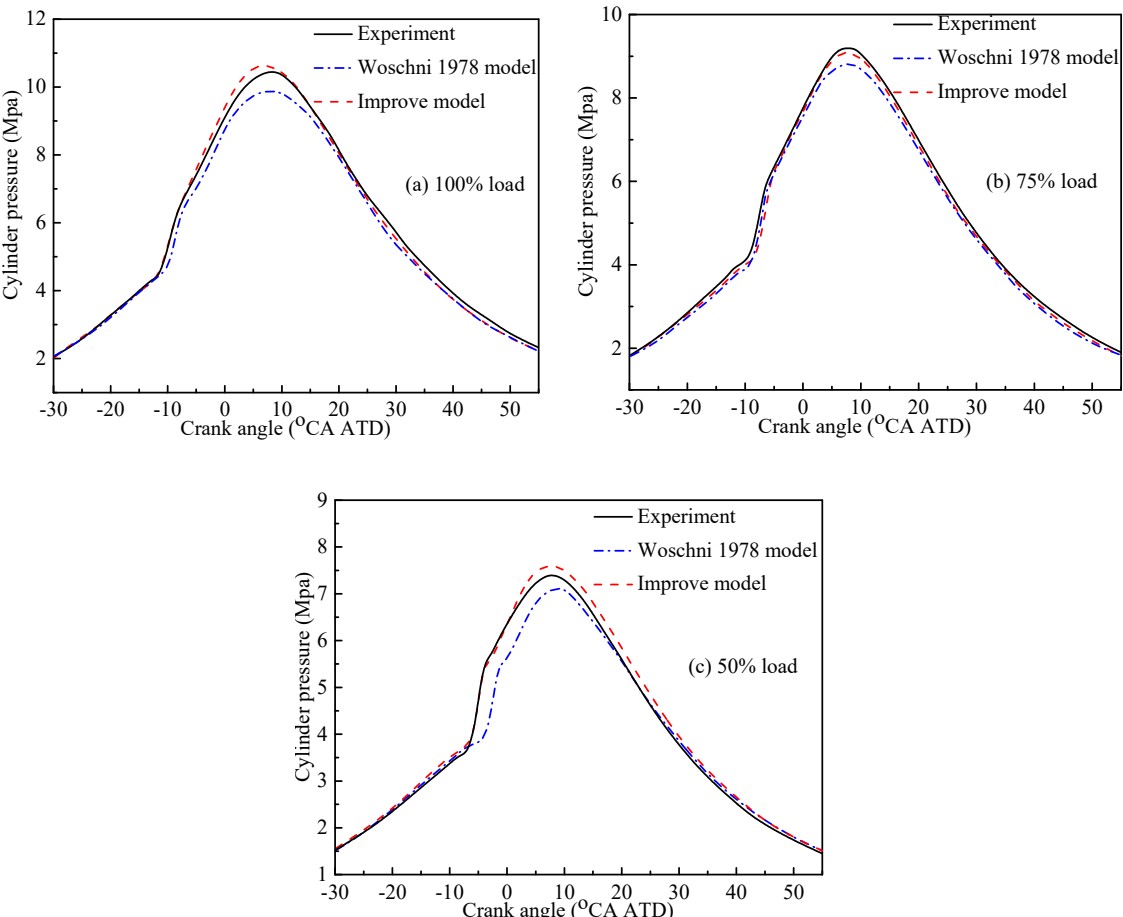

**Figure 3.** Comparison of cylinder pressure at 1000 rpm under different load conditions.

## 3. Model Application

Firstly, the experiments of diesel engines fueled with biodiesel were carried out at 1000 rpm with 10%, 25%, 50, 75% and 100 loads, respectively. Then the experiments were carried out at 628, 799, 911 and 1000 rpms, corresponding to 25, 50, 75 and 100% load, respectively. Finally, the effects of the improved model on the combustion and emission characteristics of diesel engines fueled with RME were investigated in terms of Brake power, *BSFC*, $NO_x$, CO and HC emissions.

### 3.1. Analysis on Emission and Performance Characteristics

3.1.1. Brake Specific Fuel Consumption

To evaluate the characteristic, the *BSFC* is considered an important indicator. Figure 4a shows the comparisons of engine *BSFC* under different load conditions. It can be found that the *BSFC* firstly decreases and then increases with the increase in load when the engine speed is at 1000 rpm. In addition, Figure 4b shows that with the increase in speed, the engine load also firstly decreases and then increases. For instance, the engine *BSFCs* are 247.9 and 235.3 g/(kW·h) when the load is 75% and engine speed is 911 rpm, respectively. The *BSFC* is the lowest when the engine speed is 911 rpm or the load is 75%. This is due to the fact that the operating points are the common design points. At high load, the

combustion in the cylinder becomes worse due to the insufficient oxygen. On the contrary, because of the low combustion temperature and inject pressure, the engine always gains the dropped combustion. Thus, the *BSFC* increases. In addition, it can be found the results predicted by the improved model is more accurate than those predicted by the model without considering the swirl and boiling heat transfer. For instance, the max error of improved model is 2.1%, but the maximum error of the original model is 3.1% compared with the experimental result. This is due to the fact that boiling phenomena often occur in high heat flux areas on the coolant side in the water-cooled cylinder head and the swirl will occur in the cylinder, then the heat transfer coefficient is improved. Thus, the improved model can improve the accuracy of heat transfer.

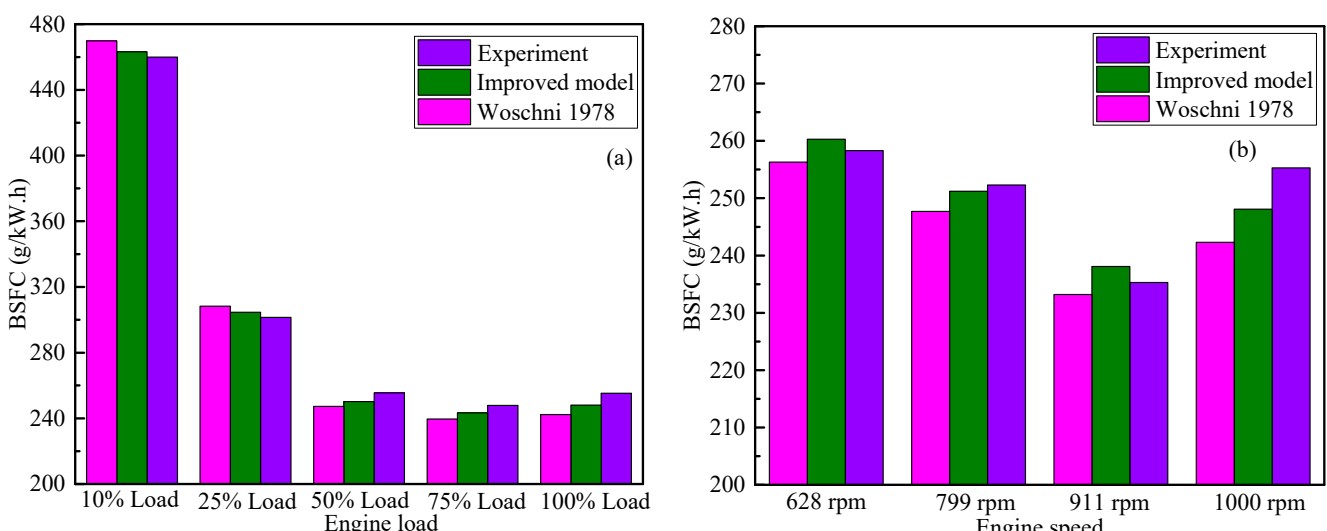

**Figure 4.** Comparison of *BSFC* under different load conditions.

### 3.1.2. Brake Thermal Efficiency

The useful power converted by the fuel combustion is defined as the brake thermal efficiency (*BTE*). The *BTE* can be calculated by the following Equation (15).

$$BTE = \frac{3600}{BSFC \cdot LCV} \tag{15}$$

where *LCV* is the fuel lower calorific value.

Figure 5 shows the comparisons of engine *BTE* under different load conditions. It can be found that the *BTE* firstly increases and then decreases with the increase in engine load (see Figure 5a) and speed (see Figure 5b). When the engine speed is 911 rpm or the load is 75%, the *BTE* is high. The experimental max. *BTE* is 37.6% when the engine speed is 1000 rpm. In addition, the experimental max. *BTE* is 38.7% when the engine speed is 911 rpm. As previously mentioned, the *BTE* is low under the other condition of diesel engines, except under the engine design conditions. As previously described, the operating points are the common design points. When the *BSFC* is high, the *BTE* is low. In addition, it can be found the values of *BTE* predicted by the improved model is more accurate than those predicted by the model without considering the swirl and boiling heat transfer. This is due to the fact that the swirl and boiling heat transfer is considered in the model. The improved model can better predict the heat transfer coefficient. Thus, the improved model can improve the accuracy of *BTE*.

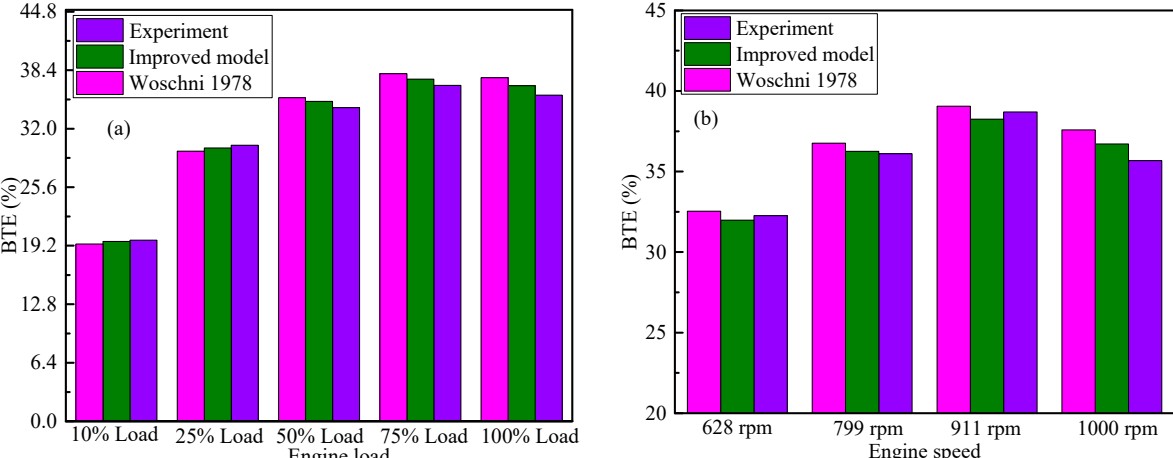

**Figure 5.** Comparison of brake thermal efficiency (*BTE*) under different load conditions.

### 3.1.3. Brake Power

Figure 6 shows the comparisons of brake power under different load conditions. It can be found that the brake power increases with the increase in engine load (see Figure 6a) and engine speed (see Figure 6b). This is due to the fact that more heat energy is produced by the increased fuel mass with an increase in engine load. In addition, it can be found brake power predicted by the improved model is more accurate than that predicted by the model without considering the swirl and boiling heat transfer. For instance, the measurement value is 193.5 kW when the engine load is 100% and the speed is 1000 rpm. The error of the improved model is 1.7% compared with the measurement value. However, the error of the original model is 2.8% compared with the measurement value. In addition, the measurement value is 20.1 kW when the engine load is 10% and the speed is 1000 rpm. The error of the improved model is 1.6% compared with the measurement value. However, the error of the original model is 2.3% compared with the measurement value. Thus, the improved model considering swirl and boiling heat transfer can improve the accuracy of heat transfer.

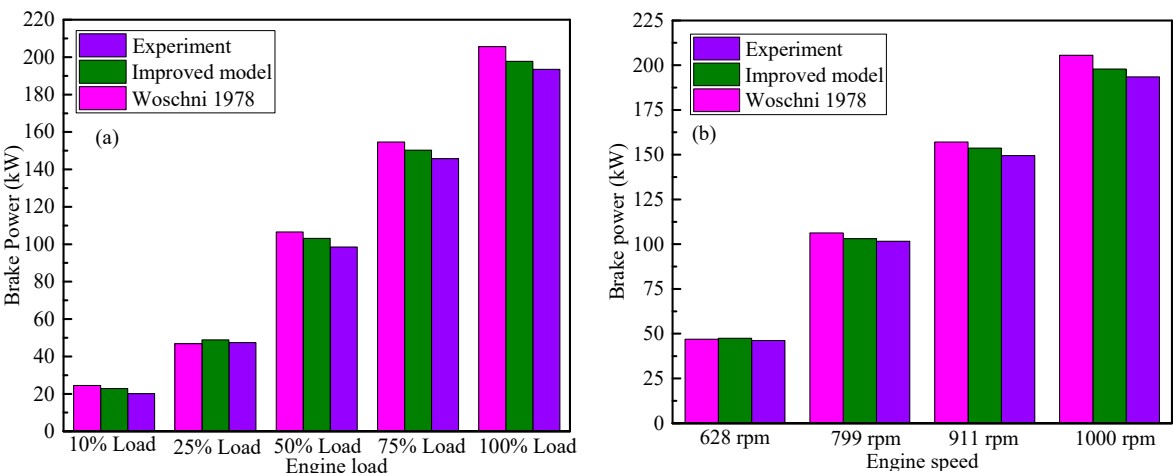

**Figure 6.** Comparisons of brake power under different load conditions.

### 3.1.4. Cylinder Temperature

The comparison of a cylinder temperature of a diesel engine fueled with biodiesel is shown in Figure 7. It can be observed that the in-cylinder temperature significantly increases with the increase in engine load and speed. This is due to the increased fuel

mass. In addition, it can be observed that the improved model can predict the cylinder temperature of diesel engines under 100% load (see Figure 7a), 75% load (see Figure 7b) and 50% load (see Figure 7c) compared to the predictive results with the experimental results. However, the improved model is more accurate. More specifically, the max. cylinder temperature predicted by the improved model is 1728 K, 1657 K and 1587 K, respectively. The corresponding errors of the improved model are 2.9%, 2.3% and 3.1% compare with the measurement value. In addition, the max. cylinder temperature predicted by the original model is 1618 K, 1548 K and 1576 K, respectively. The corresponding errors of the improved model are 3.6%, 3.5% and 3.0% compare with the measurement value. When the nucleate boiling and swirl occurs at the surface, the heat transfer in the cylinder can be calculated accurately by using the improved model. Based on the analysis above, the improved model can accurately predict the cylinder temperature of diesel engines under different conditions.

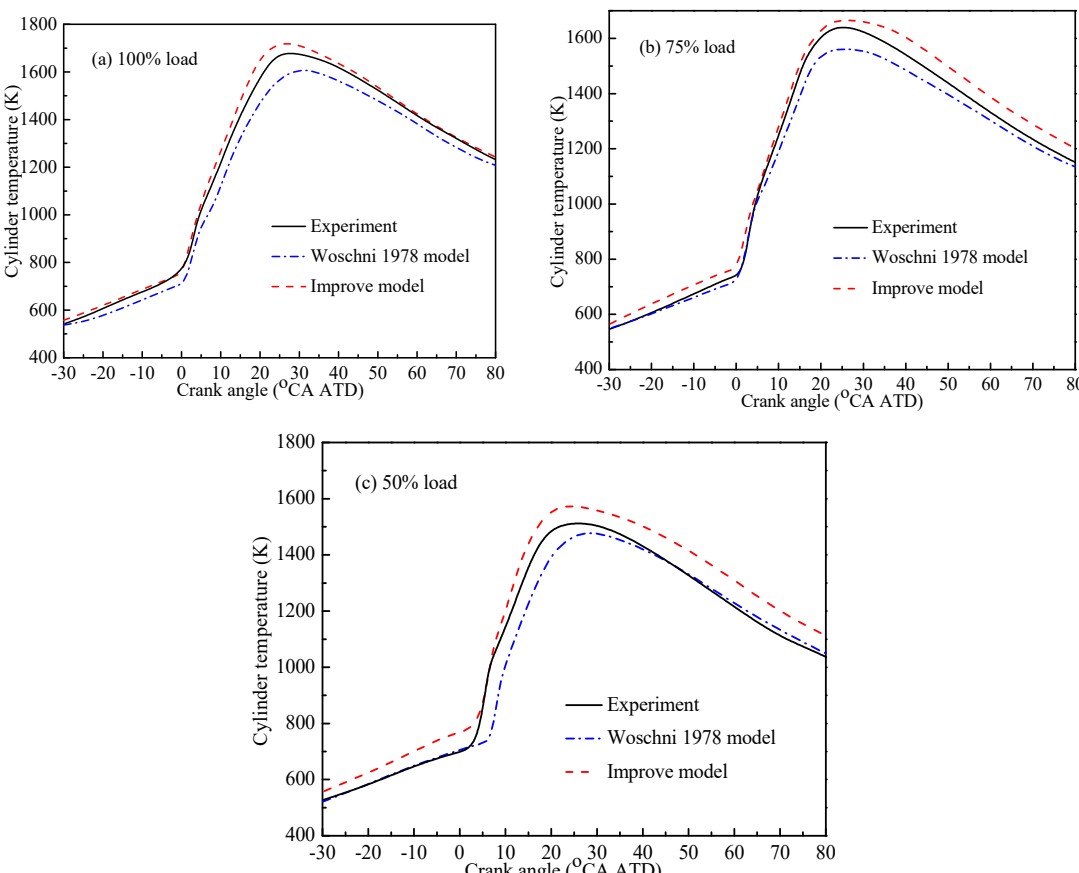

**Figure 7.** Comparison of cylinder temperature under different load conditions.

### 3.1.5. NO$_x$ Emission

The comparison of NO$_x$ emission diesel engine fueled with biodiesel is shown in Figure 8. It can be observed that with the increase in engine speed (see Figure 8a) and engine load (see Figure 8b), the NO$_x$ emission firstly decreases, then increases, and finally decreases. The temperature, oxygen concentration and reaction time are important for the information of NO$_x$. At low load, the shorter combustion duration and lower cylinder temperature result in a reduction in the reaction time. However, the increased cylinder temperature is beneficial to the formation of NO$_x$ when the engine load increases. NO$_x$ emission reaches a maximum when the load is 75% and then decreases by further increasing the load. At high load, the poor oxygen zones are formed by a lot of fuel injection sprayed into the cylinder. Thus, the NO$_x$ emission is reduced. As mentioned above, the engine

load increases gradually with the increase in engine speed. When the speed is 799 rpm, the $NO_x$ emission reaches a maximum and then decreased by further increasing the speed. In addition, it can be found that the heat transfer in the cylinder has a great influence on the cylinder temperature and the formation of $NO_x$. Similarly, the improved model predicts better the $NO_x$ formation of diesel engines under different load conditions compared with the original model. More specifically, the max. errors predicted by the improved model is 3.7% and the max. errors predicted by the original model is 4.8%. Thus, the improved model can better predict the formation of $NO_x$ and the prediction error is reduced a lot. The boiling heat transfer has a great effect on the $NO_x$ formation.

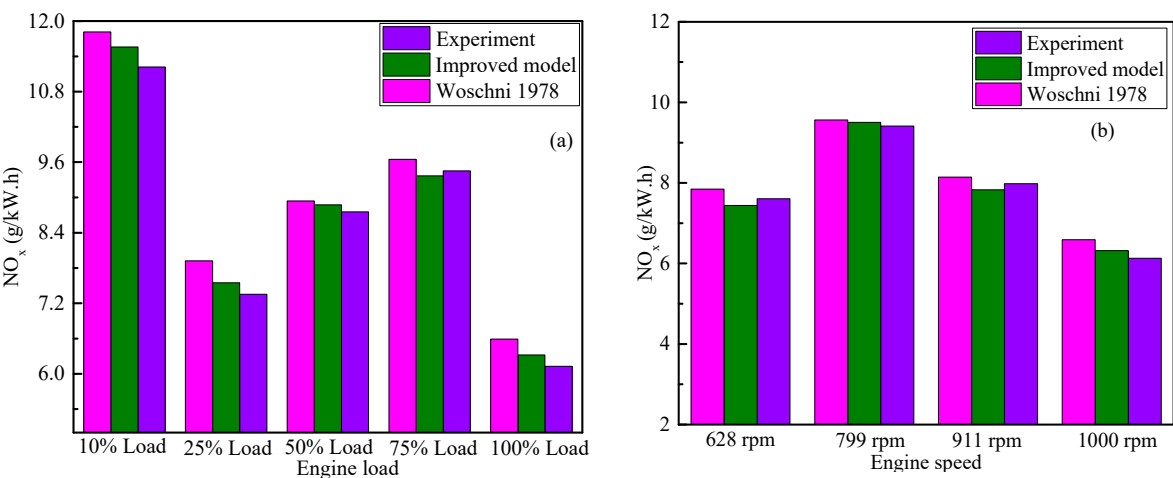

**Figure 8.** Comparisons of $NO_x$ emission under different load conditions.

The brake-specific $NO_x$ emission is an important parameter for the environmental friendliness of diesel engines. According to the Annex VI of MARPOL 73/78, the Weighting coefficient of E3 test cycle is shown in Table 4 and the brake-specific $NO_x$ emissions can be calculated by Equation (16).

$$W_{NO_x} = \frac{\sum\limits_{i=1}^{i=n} M_{NOx} \cdot W_{Fi}}{\sum\limits_{i=1}^{i=n} P_i \cdot W_{Fi}} \tag{16}$$

where $W_{NOx}$ is the brake specific emission values, the $W_{Fi}$ is the Weighting coefficient, $P_i$ is the brake power, $M_{NOx}$ is the exhaust gas mass.

The standards of Tier I, Tier II and Tier III are 11.3 g/(kW·h), 8.98 g/(kW·h) and 2.26 g/(kW·h), respectively. According to Equation (16), the $NO_x$ emission of the weighted algorithm is 8.34 g/(kW·h). Thus, brake-specific $NO_x$ emission is satisfied with the requirement of Tier II. Therefore, as the emission legislation is becoming stricter and stricter, the internal purification technology and exhaust gas treatment should be considered, and the regulation requirement is meet in future work.

**Table 4.** The Weighting Coefficient of E3 Test Cycle.

| Type | Item | Valves | | | |
|------|------|--------|---|---|---|
| | Engine speed (rpm) | 1000 | 909 | 799 | 628 |
| E3 | Load | 100% | 75% | 50% | 25% |
| | Weighting coefficient | 0.2 | 0.5 | 0.15 | 0.15 |

### 3.1.6. Hydrocarbon Emission

The hydrocarbon (HC) emission of diesel engines fueled with biodiesel under different conditions is shown in Figure 9. It can be found that the HC emission firstly decreases and

then increase with the increase in engine speed (see Figure 9a) and engine load (see Figure 9b). At low load, the low cylinder temperature is not beneficial to the oxidation of HC. With the increase of load, more HC will be oxidized due to the increasing cylinder temperature. However, the HC formation would be dominated by the impact of bad atomization caused by the high viscosity, resulting in the increase of HC emission at low load. With the further increase of load, the atomization of the fuel is gradually improved. Thus, the HC emission is reduced. Nevertheless, at high load, the HC formation would be dominated by the impact of insufficient oxygen in the cylinder resulting in the increase of HC emission. The poor oxygen content is not favorable the HC oxidation. The improved model considering the swirl and boiling heat transfer in the cylinder agrees better with the experimental results compared with the model without considering the swirl and boiling heat transfer. More specifically, the max. errors predicted by the improved model is 3.2% and the max. errors predicted by the original model is 4.1%. Base on the analysis above, the improved model improves the calculation accuracy.

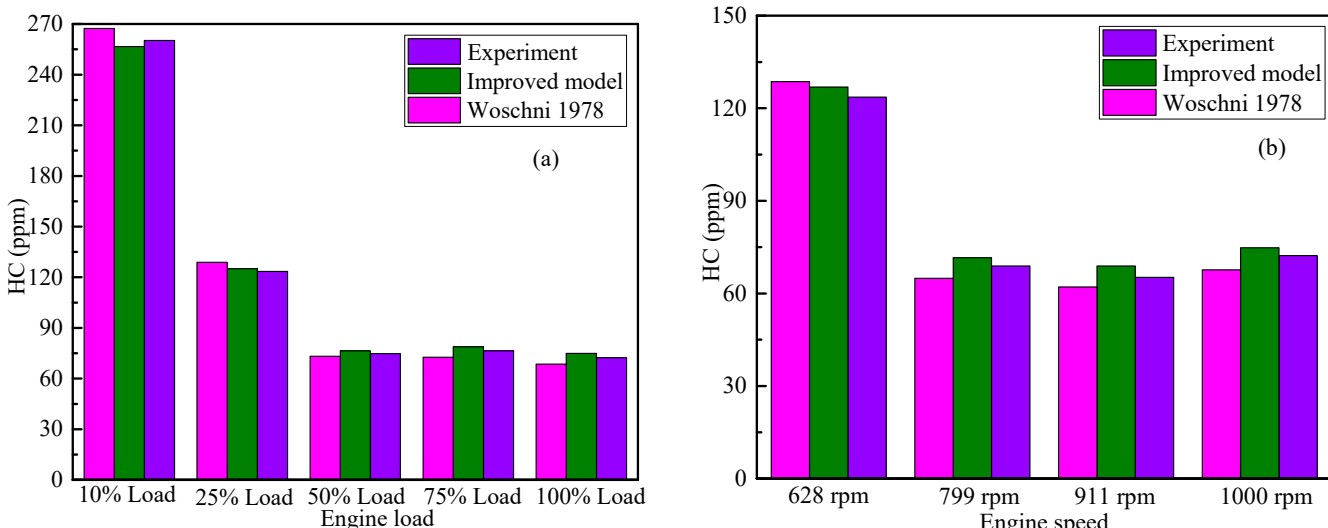

**Figure 9.** Comparisons of HC emission under different load conditions.

### 3.1.7. CO Emission

The CO is a toxic gas, and the fuel incomplete combustion will generate the CO. The oxygen content in biodiesel can promote the oxidation of CO to $CO_2$. Thus, biodiesel can decrease harmful gas emissions. The comparisons of CO emission under different conditions are shown in Figure 10. It can be observed that with the increase in engine speed (see Figure 10a) and engine load (see Figure 10b), the CO emission firstly decreases and then increases. At low load, the low cylinder temperature is not beneficial to the oxidation of CO to $CO_2$. With the increase in load, more CO will be oxidized due to the increasing cylinder temperature. Nevertheless, with the further increase in load, the CO formation would be dominated by the impact of insufficient oxygen in the cylinder resulting in an increase in CO emission. The improved model considering the swirl and boiling heat transfer in the cylinder agrees better with the experimental results compared with the model without considering the swirl and boiling heat transfer. More specifically, the max. errors predicted by the improved model is 3.1% and the max. errors predicted by the original model is 3.6%. Based on the analysis above, the influences of boiling heat transfer and swirl on diesel engines are very important.

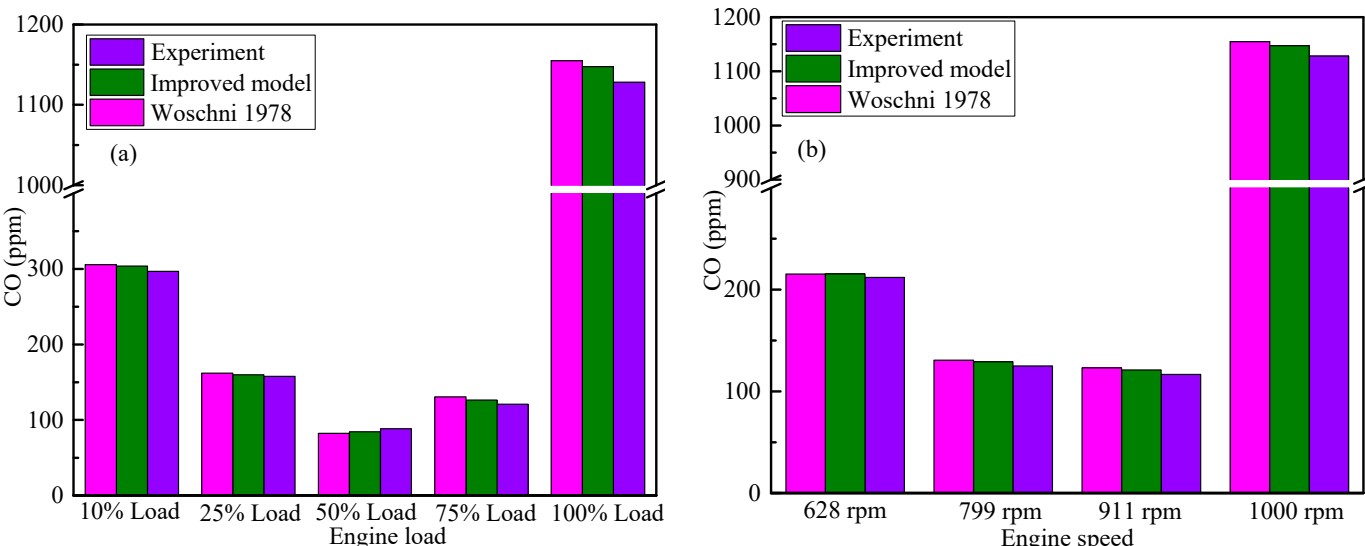

**Figure 10.** Comparisons of CO emission under different load conditions.

### 3.2. Input Parameter Analysis

The thermodynamic cycle of diesel engines is extremely complex. The influences of input parameters on the output of diesel engines are different. Thus, the perturbation method is very meaningful to investigate the relative contribution of every input on the performance and emission characteristics of diesel engines. In the study, an improved version of perturbation is developed. The relative effects of different parameters such as *EGR*, injection mass, injection timing, compression ratio, inlet air pressure, fuel injection pressure, fuel injection duration and inlet air temperature on performance and emission characteristics are compared.

The perturbation method is a better way to investigate the relatively important inputs as the complex nonlinear function with a lot of output data and input data. The required data are provided by using the AVL-BOOST model. Thus, 3600 operational points are generated by the improved model in the AVL-BOOST environment. The purpose is to find the relative important input factors on the different outputs of diesel engines. This formula with 8 pre-defined and 4 outputs factors was established.

$$Y_{4\times1} = f(X_{8\times1}) \tag{17}$$

where $Y_{4\times1}$ and $X_{8\times1}$ are the vectors of output and input, respectively.

The upper and lower bounds for data generation are defined in Table 5. Based on the perturbation theory, the relative important input factors can be calculated by the following Equation(18):

$$\overline{s_{ij}} = \overline{\frac{\partial f}{\partial x}} \times \overline{\frac{\sigma_J}{\sigma_I}} = \frac{\sum_{k=1}^{n} \frac{f(X, x_{j,k}+\sigma_k) - f(X, x_{j,k})}{\sigma_k} \times \frac{\sigma_j}{\sigma_i}}{n} \tag{18}$$

where $\overline{s_{ij}}$ is the average sensitivity factor of the $j$th output date relative to $i$th input data, $\sigma_j$ and $\sigma_i$ are the standard deviations, $n$ is the divided equal intervals in the input span. A random point is selected in the defined rank. $\sigma_k$ is the perturbation of $k$th interval and should be very small to keep in the respective interval.

**Table 5.** The Upper and Lower Bounds for Data Generation.

| Parameters | Unit | Upper Bound | Lower Bound |
|---|---|---|---|
| Engine speed | rpm | 1000 | 400 |
| Equivalence ratio | - | 1.2 | 0.2 |
| Inlet manifold temperature | °C | 45 | 0 |
| Inlet manifold pressure | bar | 2.5 | 1.0 |
| *EGR* | - | 0.4 | 0 |
| Injection mass | g | 1.32 | 0.04 |
| Injection timing | °CA | −30 | −10 |
| Exhaust pressure | bar | 3 | 0.8 |
| Fuel injection pressure | bar | 150 | 50 |
| Compression ratio | - | 19 | 10 |

The engine speed and equivalence ratio can be considered as the important factors affecting the engine characteristic. It is very interesting to investigate the parameter contributions on the output parameters under different conditions. The values of the *y*-axis in Figures 11–14 show the relative contrition of each input on different output parameters. The relative contrition is the dimensionless quantity. The positive contribution shows the increase in the input parameter will cause an increase in the output performance index. On the contrary, the negative contribution shows that the increase in the input parameter will result in a reduction in the output performance index. Four output parameters are power, *BSFC*, $NO_x$ and HC. In Figures 11–14, the numbers 1–8 represent *EGR*, Compression ratio, Exhaust pressure, Fuel injection pressure, Injection mass, Injection timing, Inlet air pressure and Inlet air temperature.

As can be seen in Figures 11–14, the fuel injection mass per cycle has the most efficient contribution in almost all the output performances and is affected by injected fuel mass, more than others. Its effect improves in leaner mixtures. Additionally, inlet pressure is an effective parameter on brake power. However, the effects of inlet pressure increase on power increase are not comparable to injected fuel mass increase but its effect enhances in less lean regimes.

The *BSFC* is more than anything affected by inlet pressure. The increase in inlet pressure will improve the combustion and reduces the *BSFC*. This is due to its effects on pumping losses. Its effects are dominant in high engine speeds. Additionally, the increase in fuel mass can decrease *BSFC* in leaner mixtures due to its effects on power increase. In addition, the increases in compression ratio, injection pressure and injection timing will reduce the *BSFC*.

The $NO_x$ generation is influenced by the *EGR* rate more than other parameters while it will increase the soot generation, especially in less lean mixtures. Similarly, the *EGR* rate effects are more in lower engine speeds. The injection timing is the other parameter that can affect the $NO_x$ generation. Retarding the injection improves the $NO_x$ generation. In addition, the fuel injection mass per cycle will increase the $NO_x$ generation; also increasing the fuel injection pressure will increase the $NO_x$ generation especially in less lean operation conditions.

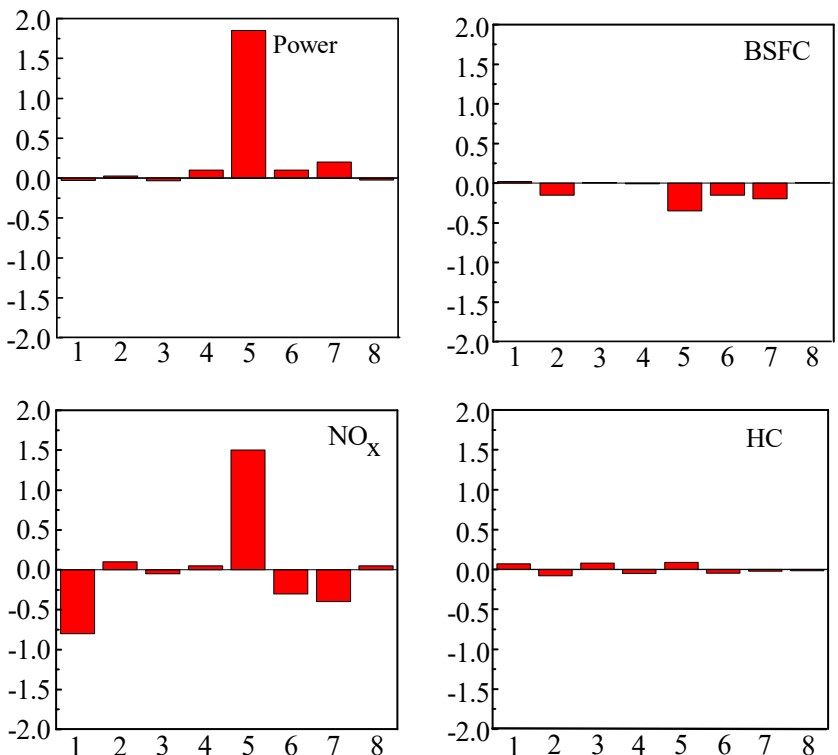

**Figure 11.** Comparisons of relative contribution on the engine characteristic in low speed (628 rpm) and in lean operation zone (Equivalence ratio $\Phi = 0.25$).

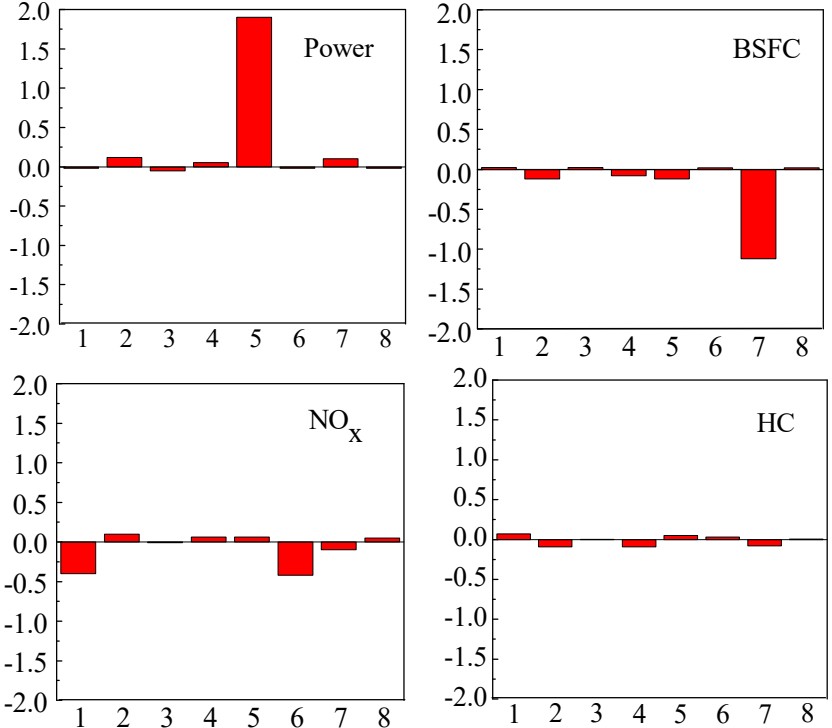

**Figure 12.** Comparisons of relative contribution on the engine characteristic in high speed (1000 rpm) and in lean operation zone (Equivalence ratio $\Phi = 0.25$).

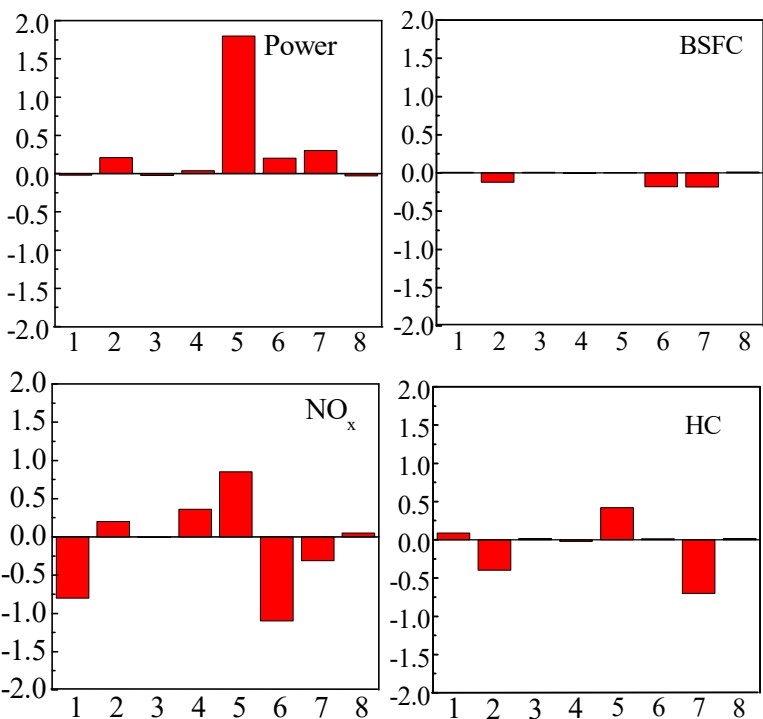

**Figure 13.** Comparisons of relative contribution on the engine characteristic in low speed (628 rpm) and in lean operation zone (Equivalence ratio $\Phi = 0.75$).

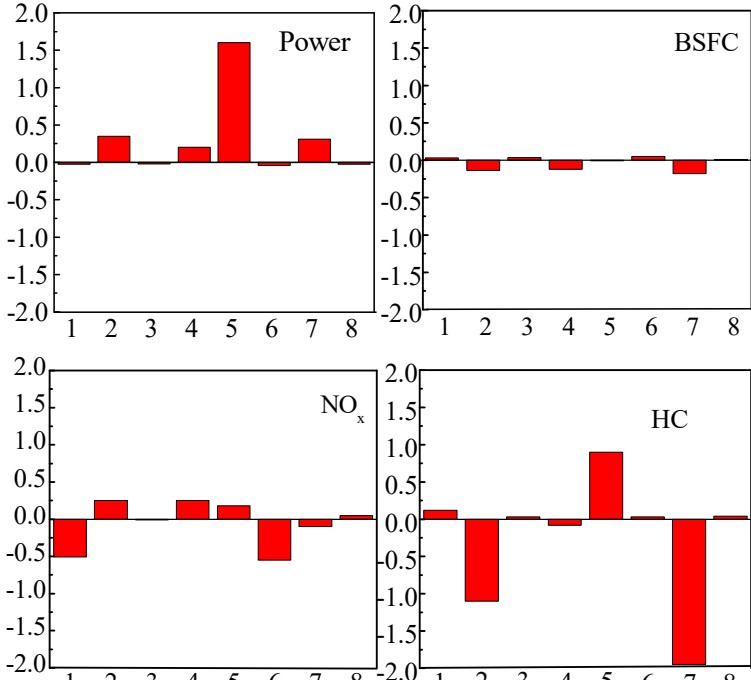

**Figure 14.** Comparisons of relative contribution on the engine characteristic in in high speed (1000 rpm) and lean operation zone (Equivalence ratio $\Phi = 0.75$).

As it is expected, the increase in the fuel injection mass per cycle will increase the HC generation in whole engine operating conditions but low engine speed and very lean condition in which HC generation improves by increasing the fuel mass increasing. The increase in the *EGR* rate will increase the HC generation. Its effects are gained in very lean operation as a lower temperature is obtained in the cylinder and the rate of HC oxidization

decreases. In addition, the increase in inlet pressure can also decrease the HC generation in less lean conditions. The increase in fuel injection pressure can decrease HC generation in high engine speeds. This is due to the fact that the high fuel injection pressure can decrease the duration of fuel injection and increase the maximum cylinder pressure.

## 4. Conclusions

Today the environmental pollution [43–51] and energy crisis [52–58] have to learn another new problem. The thermodynamic process of the diesel engine is a complicated process inside the cylinder. To evaluate the practicability of biodiesel, an improved heat transfer model is proposed and employed to investigate the combustion and emission characteristics of diesel engines. The improved heat transfer model mainly considers the influence of the swirl and boiling heat transfer inside the engine. In addition, a chemical kinetics mechanism including 475 reactions and 134 species is employed to predict the combustion. Finally, a perturbation method is employed to investigate the relatively important inputs as the complex nonlinear function with lots of output data and input data. The main conclusions are summarized as follows:

(1) The results predicted by the improved model are more accurate than those predicted by the model without considering the swirl and boiling heat transfer. The max errors of *BSFC*, brake power, cylinder temperature, $NO_x$ emission, HC emission and CO emission predicted by the improved model are 2.3%, 1.7%,3.1%, 3.7%, 3.2 and 3.1%, but the maximum errors of the original model are 2.1%, 2.8%, 3.6%, 4.8%, 4.1% and 3.6% respectively compared with the experimental result.

(2) The oxygen content of biodiesel is favorable for the oxidation of HC and the formation of $NO_x$. However, the effect of HC formation would be dominated by the impact of bad atomization caused by the high viscosity, resulting in the increase in HC at low load.

(3) The fuel injection mass plays an important role in emission and performance characteristics. In addition, the *EGR*, compression ratio and inlet air pressure have a great effect on the $NO_x$ emission. The increase in inlet pressure and compression ratio will decrease the *BSFC*. The inlet air pressure, compression ratio and fuel injection pressure can improve the fuel combustion and reduce the HC emission. In the case of the same operation input parameters, the contribution of operation input on the output is not independent, but the output is changed by changing other parameters. Therefore, it needs to be further improved in future work.

**Author Contributions:** D.T.: Conceptualization, methodology, resources, project administration, writing—original draft preparation; Z.C.: software, formal analysis, writing and editing; J.L. (Jiangtao Li): software, investigation; J.L. (Jianbin Luo): project administration, writing—review and editing; D.Y.: writing—review and editing; S.C.: writing—review and editing; Z.Z.: project administration, writing—review and editing, supervision. All authors have read and agreed to the published version of the manuscript.

**Funding:** This work is supported by the Natural Science Foundation of Fujian under the research grants of 2020J01690 and this work is supported by the cultivated Natural Science Foundation of Jimei University under the research grants of ZP2021008, the Science and Technology Project of Guangxi, China (GK AD19245149) and the Science and Technology Project of Liuzhou, China (2019DH10601).

**Data Availability Statement:** All data used to support the findings of this study are included within the article.

**Conflicts of Interest:** The authors declare no conflict of interest.

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
