# Peer review of "Effects of Swirl and Boiling Heat Transfer on the Performance Enhancement and Emission Reduction for a Medium Diesel Engine Fueled with Biodiesel"

_processes, doi:10.3390/pr9030568_

Round 1

Reviewer 1 Report

Comments and suggestions are detailed in the attached file.

Reviewer 2 Report

Summary of the review:

In this study, the author attempts to investigate the effects of including swirl and boiling heat transfer in a model developed by using AVL BOOST software, using RME biodiesel. It is nowhere mentioned and I assume that the author used a blend of rapeseed oil methyl ester (RME) and diesel, the proportions of which are also unknown. One of the previous studies referenced by the author for details (reference 27) does talk about the blend proportions (the same engine schematic diagram is being used in this study from reference 27), however those results using the biofuel have been reported earlier. Moreover, this study should be a stand alone paper with all information provided, even if the same experimental data or set up was used.

The paper lacks novelty. Currently, the author has based this paper on including swirl and boiling heat transfer in the AVL BOOST model. Adding swirl and boiling heat transfer to the model makes the model agreeable with the experimental results. There are a fair few coefficients which can be modified to fine tune the results in AVL BOOST software. The author has used motoring tests to find those coefficients. Investigations based on effect of air swirl and boiling heat transfer in the hot spots have been attempted in previous studies and therefore including those in a model (where the coefficients can be fine tuned) does not bring the required novelty in this study.

On the other hand, the use of biofuel is only secondary and has not been included as the primary novel characteristic of this study, most probably because the results from using the biofuel have been reported in reference 27 before.

The relative contribution of each variable such as EGR, injection mass, injection timing, etc on the performance and emissions of the diesel engine on certain speeds and eq ratio conditions were also investigated by the perturbation theory using the AVL BOOST software. Those results are interesting, however they also do not contribute to any novel knowledge.

My recommendation for this paper is that although the results are interesting, they do not make a novel contribution to knowledge about the diesel engine combustion and emissions. Currently, the paper does not qualify to be published in a journal due to lack of novelty.

Some major and minor issues that have been observed in the review are below.

Major and minor issues:

Major proof reading is required. There are numerous grammar errors in the manuscript. I suggest getting it proof-read by a professional proof-reader or somebody expert in the English language. For example, line 16, delete “in” (in inside the engine), line 21, change “lots of”, same line, it should be “improved”. Line 32, “trucks”. Similarly, sentence formation at many places needs to be improved.

Line 45, “open closing”

Line 48, remove “the” the shorter engine start-up time.

Line 55, change to “a lot of money and time”.

Line 74, “Considered”

Check at other places too. At present the language of the manuscript and grammar is at the stage that it is hard to follow the content.

Line 75, I think you want to write “nucleate boiling”, rather than nuclear.

The introduction section is quite poorly written, there is no flow. The information has to flow from a broad to narrow perspective, ending at the research gap. Currently, the information is jumping here and there without a properly justified research gap. The novelty of the study is not discussed properly. There is no discussion about the biofuel novelty. It should be discussed why this particular biofuel is used if the novelty is knitted around that. Adding swirl and boiling heat transfer to the AVL model does not justify the novelty of the study. I suggest the introduction section should be restructured and a justified research gap should come in the last paragraph.

Line 103, a reference should be used to state from where Equation 1 has been used.

Similarly, references for Equation 2, 3, 4, 5, 6, 7, 8, 9.

Usually, units are expressed in italics.

Line 119, “known” Line 127, “accordance”.

Line 221, why have these speeds (628, 799, 911 and 1000 rpm) been chosen.

There is no depth of explanation whatsoever in section 3.1.1 (BSFC)

Line 298 and 367 “based”

Line 306-307 should be referenced.

Section 3.1.1 and 4.1.2 – there is no depth in the discussion. The observations of the graph are reported as it is. It must include why the BSFC or BTE is higher or lower at a particular load or condition.

Same section number 4.1.6 used for CO and HC.

In Section 4.1.5 and 4.1.6 (NOx and HC), the only explanation about NOx and HC increasing with increased load (100% load) is insufficient oxygen. The explanation about CO decreasing with load is increased in-cylinder temperature. The engine behaviour at 100% load is significantly different and it does not have an in-depth explanation. Does it have to do anything with the biofuel? All the depth in the discussion is completely missing.

Figure 9 – the y axis says CO, it should be HC.

Numbers 1-8 on the x-axis should be mentioned either in the graph or in the figure caption in Figures 11-14. The depth of the discussion on why a certain engine parameter has lower or higher effect on the engine performance and emissions is missing at the moment.

Reviewer 3 Report

After reviewing the manuscript, I have some comments as follows:

  1. In this study, the authors simulated the combustion and emission formations of a diesel engine fueled with biodiesel by using the AVL-BOOST code with an improvement in the heat transfer model by Authors. The Authors conducted a number of simulations and experiments, and have successfully investigated the combustion and emission characteristics of the engine at different engine speeds and loads. The simulation results using the improved heat transfer model showed good agreements with the simulation results using Woschni 1978 model as well as the experimental results. However, in subsection 3.1, the Authors just compared the results between the two models and experiments as well as presented the tendencies in combustion properties and emission characteristics according to engine loads and speeds but didn’t explain the reasons why there were those such trends. Therefore, in the revised manuscript, the Authors should add more explanations for why there were the trends as having been presented.
  2. Regarding the chemical kinetics mechanism. The Authors mentioned employing a chemical kinetics mechanism including 475 reactions and 134 species to predict the combustion and emissions of the engine but didn’t clarify if the mechanism was built by the Authors or where was it from. The Author needs to clarify this.
  3. In subsection 4.2 (Input parameter analysis), the Authors presented the influences of input parameters on the output parameters of the engine. However, in each case of the engine’s operating conditions (engine speeds and equivalence ratios – presented by Fig. 11 - 14), the Authors just indicated how much the input parameters can affect output parameters as well as the effecting tendencies of input parameters on output parameters without explaining why those input parameters affect the output parameters like that. For instance, Fig. 11 shows that the increase in EGR leads to a decrease in NO and an increase in HC, but the authors do not explain why there are such trends. Therefore, it is better if the Authors add explanations for the trends. Additionally, figures 11 – 14 also show that at the same engine speed but with different equivalence ratios (or the same equivalence ratio but with different speeds) the effecting intensities and/or tendencies of input parameters on output parameters are different. Therefore, the Authors should supplement to explain why there are these such differences.
  4. There are some errors in English spelling and grammar need to be checked and corrected.
  5. There is an error in numbering the subsection (3.1.1 to 4.1.2 (must be 3.1.2)). Additionally, the explanation for Fig. 12 was wrong and need to be corrected.

Round 2

Reviewer 1 Report

The response provided by the authors for the Question 5 is very useful but it doesn´t clarify if the model proposed is based on Woshcni one.

The tilte of the table 1 should  be: “Properties and composition of RME”.

Which are the units of y-axis in Figures 11-14

Reviewer 2 Report

The authors have done extensive revision of the manuscript since the last review. The concerns raised in the previous review have been addressed thoroughly by the author. However, please double check if the reference table follow the Applied Sciences guidelines and all references are consistent.
